

# Determination of phase space region for cross-check validation of the neutron detection in the BINA experiments

B. Włoch[1⋆], K. Bodek[2], I Ciepał[1], M. Eslami-Kalantari[3], J. Golak[2], N. Kalantar-Nayestanaki[4], G. Khatri[5], St. Kistryn[2], B. Kłos[6], A. Kozela[1], J. Kuboś[1], P. Kulessa[7], A. Łobejko[6], A. Magiera[2], H. Mardanpour[4], J. Messchendorp[4], I. Mazumdar[8], W. Parol[1], A. Ramazani-Moghaddam-Arani[4], D. Rozpędzik[2], R. Skibiński[2], I. Skwira-Chalot[9], E. Stephan[6], A. Wilczek[6], H. Witała[2], A Wrońska[2] and J. Zejma[2]

**1** Institute of Nuclear Physics Polish Academy of Sciences, PL-31342 Krakow, Poland
**2** M. Smoluchowski Institute of Physics, Jagiellonian University, PL-30348 Krakow, Poland
**3** Department of Physics, School of Science, Yazd University, Yazd, Iran
**4** KVI-CART, University of Groningen, NL-9747 Groningen, The Netherlands
**5** CERN, CH-1211 Geneva, Switzerland
**6** Institute of Physics, University of Silesia, PL-41500 Chorzów, Poland
**7** Forschungszentrum Jülich, Institut für Kernphysik, D-52428 Jülich, Germany
**8** Tata Institute of Fundamental Research, 400 005 Mumbai, India
**9** Faculty of Physics University of Warsaw, PL-02093 Warsaw, Poland

⋆ boguslaw.wloch@ifj.edu.pl

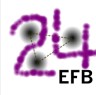

## Abstract

Deuteron breakup reactions are basic laboratories for testing nuclear force models. Recent improvements in the data analysis allow for direct identification of neutrons in the BINA detection setup. This opens up the opportunity to study new aspects of few-nucleon system dynamics like charge dependence of nuclear force or Coulomb interaction. In this paper we determine regions along the kinematical curves where differential cross section of deuteron-proton breakup reactions can be measured by the proton-neutron and proton-proton coincidences simultaneously. This is particularly useful for validation of the neutron detection technique.


## 1 Introduction

Investigations of few-nucleon systems gives suitable testing ground for modern nuclear interaction models. The simplest system, in which different theoretical models can be tested, are those composed of three nucleons (3N). The differential cross sections of the deuteron breakup

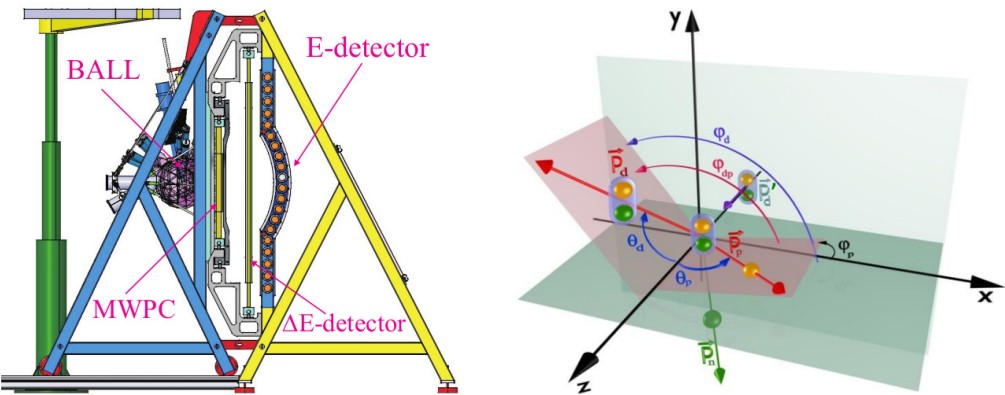

Figure 1: *Left panel:* A side view of the BINA experimental setup. *Right panel:* A schematic view of the breakup reaction with indicated polar ($\theta_p$), azimuthal ($\phi_p$) and the relative azimuthal ($\Delta\phi_{pp}$) angles of the two detected particles in the laboratory reference frame. Figures adapted from [15].

reaction are very sensitive to various aspects of the system dynamics, not only to the effects of the three nucleon force (3NF), but also to the long-range Coulomb interaction and to its interplay with strong forces. Exact solutions of the Faddeev equations with any modern nuclear potentials like the CD Bonn [1], Argonne V18 [2] or Nijmegen I and II [3] combined with models of 3NF like the Tucson-Melbourne [4] or the Urbana IX [5] are available. There are also alternative methods of construction of the potentials like the coupled channel approach [6] and, most importantly, the ones based on Chiral Effective Field Theory [7].

A new generation experiments devoted to studies of few-nucleon systems were carried out at KVI in Groningen (the Netherlands) with the use of the BINA detector and with the deuteron beam provided by the AGOR cyclotron [8]. A large set of high precision data of $^2$H($d,dp$)$n$ [9] and $^1$H($d,pp$)$n$ [10] deuteron breakup reaction were obtained at a beam energy of 160 MeV. The experiment was a continuation of previous very successful studies [11,12]. In this analysis we are focused on two kinds of the deuteron-proton breakup reactions, one with the detection of two protons and the other with proton and neutron detected in the BINA.

## 2 Experimental Setup

The BINA detection system is characterized by a large angular acceptance. It has been specially designed for investigations of few-nucleon systems in the intermediate energy range. It is composed of two main parts, the Forward Wall covering polar angles from 16° to 40° and the Backward Ball covering range from 40° to 165°. The Backward Ball consists of 149 triangular phoswich detectors arranged in a fullerene-like shape. It also plays the role of a vacuum-tight scattering chamber with a liquid deuterium or hydrogen target inside. The Forward Wall is composed of a multi-wire proportional chamber (MWPC), 12 vertical strips made of 2 mm thin plastic scintillator stripes ($\Delta E$) and 10 long and 12 cm thick horizontally arranged, plastic scintillator slabs forming a stopping detector ($E$). The vertical and the horizontal scintillators form an array of virtual $\Delta E$-$E$ telescopes used for particle identification. Detailed information about the detector can be found e.g.in [13,14].

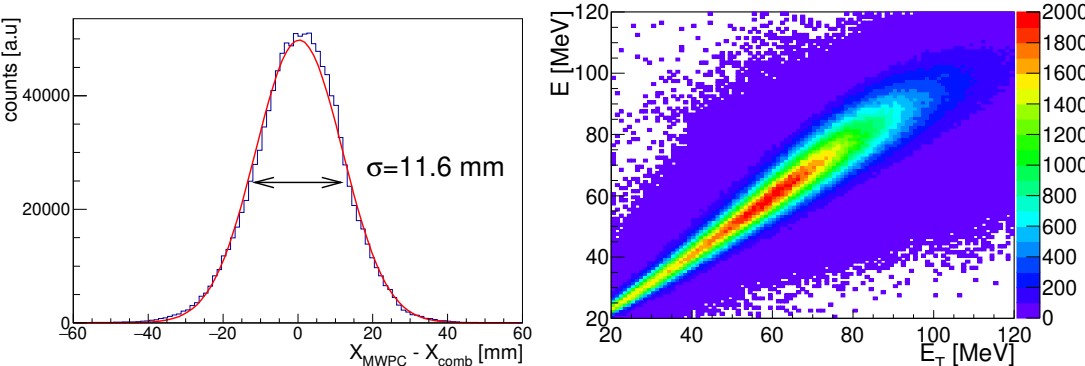

Figure 2: Check of momentum reconstruction methods on the basis of registered protons: *Left panel:* Histogram of the differences between the *x*-coordinate reconstructed from combined TDC and ADC method and the *x*-coordinate obtained from MWPC. *Right panel:* Energy reconstructed from the deposited energy in the *E* detector versus energy obtained from the time of flight method $E_T$ [19].

## 3 Neutron detection

For the neutron detection we use the Forward Wall. Among all the elements of the BINA setup, the thick *E* scintillators have the highest efficiency for neutron detection, estimated at around 10% [16, 17] which is consistent with the other experiments with a similar detection geometry and scintillating material [18]. In our detection technique, the $\Delta E$ detector and MWPC work as an active veto, rejecting possible events with charged particles except one proton necessary to identify the reaction. Signals in each thick E bar were read out from both ends by photomultipliers, allowing to obtain the information on the position.

The position of the detected neutron along the horizontal ("*x*") axis can be determined with two different methods. A TDC method is based on the time asymmetry between left and right photomultipliers while an ADC method uses the difference in the pulse height. Both methods provide independent information on the particle position and combined together result in position reconstruction with resolution of $\sigma_x =11.6$ mm (see Fig. 2). Unfortunately, these methods cannot be used for a position reconstruction along the vertical axis. Instead, we use the granularity of the *E* detector. The geometrical center of the scintillator gives a rough position in the *y*-axis with a maximal error of about 40 mm.

Taking the sum of the left and the right TDC signals removes the dependence on the hit position and allows to calculate time of flight (TOF) for a given particle. Using this information, the energy of the particle can be directly calculated. Fig. 2 illustrates the resolution of the energy reconstruction for protons with the use of TOF. To find TOF of a neutron, at least one accompanying charged particle must be registered in the Wall detector in order to calculate the time $T_0$ of the reaction. Having the neutron position and energy reconstructed, one can calculate its momentum.

## 4 Proton-neutron cross-check regions

A commonly used representation of experimental data in three-body breakup reaction is a 5-fold differential cross section. The definition of angles is shown in Fig. 1. The three particles in the final state can be described with 9 variables. The energy and momentum conservation

laws and the axial symmetry of the reaction reduces this number to 5 variables, e.g., $E_{p1}$, $E_{p2}$, $\theta_{p1}$, $\theta_{p2}$, $\Delta\phi = \phi_{p1} - \phi_{p2}$. The $E_{p1}$ and $E_{p2}$ are related by the formula:

$$E_{p1} + E_{p2} - \sqrt{2E_{beam}E_{p1}}\cos\theta_{p1} - \sqrt{2E_{beam}E_{p2}}\cos\theta_{p2} + \sqrt{E_{p1}E_{p2}}\cos\theta_{p1p2} = \frac{Q - E_{beam}}{2}, \quad (1)$$

where: $\cos\theta_{p1p2} = \cos\theta_{p1}\cos\theta_{p2} + \sin\theta_{p1}\sin\theta_{p2}\cos(\phi_{p1} - \phi_{p2})$, $E_{beam}$ is the beam (deuteron) energy, $E_{p1}$, $\theta_{p1}$ are the energy and polar angle of the first proton, $E_{p2}$, $\theta_{p2}$ are the energy and polar angle of the second proton and $Q$ is the binding energy of the deuteron.

For a chosen angular configuration ($\theta_{p1}$, $\theta_{p2}$, $\Delta\phi$) the relation between energy of the two particles defines a so-called "kinematical curve" or kinematics. Examples of kinematics associated with different angular selections are presented in Fig. 3. In order to present the cross section distributions and to avoid the confusion about which particle is the "first" and which is the "second" one, one defines the $S$-variable, an arc-length along the kinematics. It is clear that complete breakup kinematics can be reconstructed based on the knowledge of three momentum components of one particle and emission angles of the second particle.

To determine the regions of interest where both kinematics of $^1$H($d$, $pp$)$n$ and $^1$H($d$, $pn$)$p$ overlap, the following procedure has been developed based on theoretical cross sections. A set of theoretical cross sections based on the CDBonn potential [20] for both mentioned reactions has been calculated. The energy thresholds for a detection of protons (20 MeV) and neutrons (5 MeV) in the BINA setup have been introduced. For each analyzed proton-proton angular configuration, the energy and emission angles of the corresponding neutron were calculated for every point ($E_{p1}$, $E_{p2}$) on the proton-proton kinematics. If the calculated neutron angles are within the acceptance of the detector (Wall) and its energy exceeds the detection threshold, then such a point is considered as valid for the cross-check analysis. For each such point, ($E_n$, $E_p$), the position on the proton-neutron kinematics was determined. Having these regions identified, one can calculate the corresponding values of the cross sections for $^1$H($d$, $pp$)$n$ and $^1$H($d$, $pn$)$p$ reactions, respectively. Examples of the obtained distributions are shown in Fig. 4. This procedure allows for direct cross-check the analysis based on the proton-proton and proton-neutron coincidences.

## 5   Conclusion

The newly developed neutron detection method is very promising. In order to check our detection method, a simple analysis of kinematic relationship of deuteron-proton breakup reaction has been performed. The phase-space regions characterized by the possibility to extract the cross section independently from neutron-proton and proton-proton coincidences has been found. The analysis allowed to select 178 experimental points that can be used for a cross-check within the acceptance of our detection setup. In the next step, this procedure will be applied to the data. By comparing the final cross section calculated from neutron-proton and corresponding proton-proton coincidences one will be able to study systematic uncertainties of the most important quantities used for analysis involving neutrons (detection efficiency, configurational efficiency, energy reconstruction etc.). The results will be used in further studies of the deuteron breakup reaction.

## Acknowledgements

The numerical calculations were partially performed on the supercomputer cluster of the JSC, Jülich, Germany.

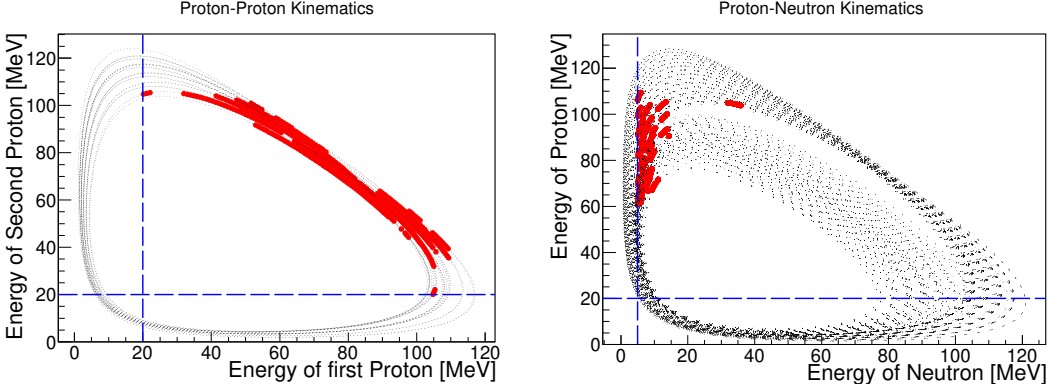

Figure 3: *Left panel:* A set of proton-proton kinematics for various combinations of $\theta_{p1}$, $\theta_{p2}$ and $\Delta\phi$ angles (typical values for these angles are shown in Fig. 4). Red lines represent the regions, where the corresponding neutron can be detected in the BINA setup together with the proton-proton coincidences. Blue dashed lines refer to applied detection thresholds. *Right panel:* The same regions as in the left panel but for the neutron-proton kinematics.

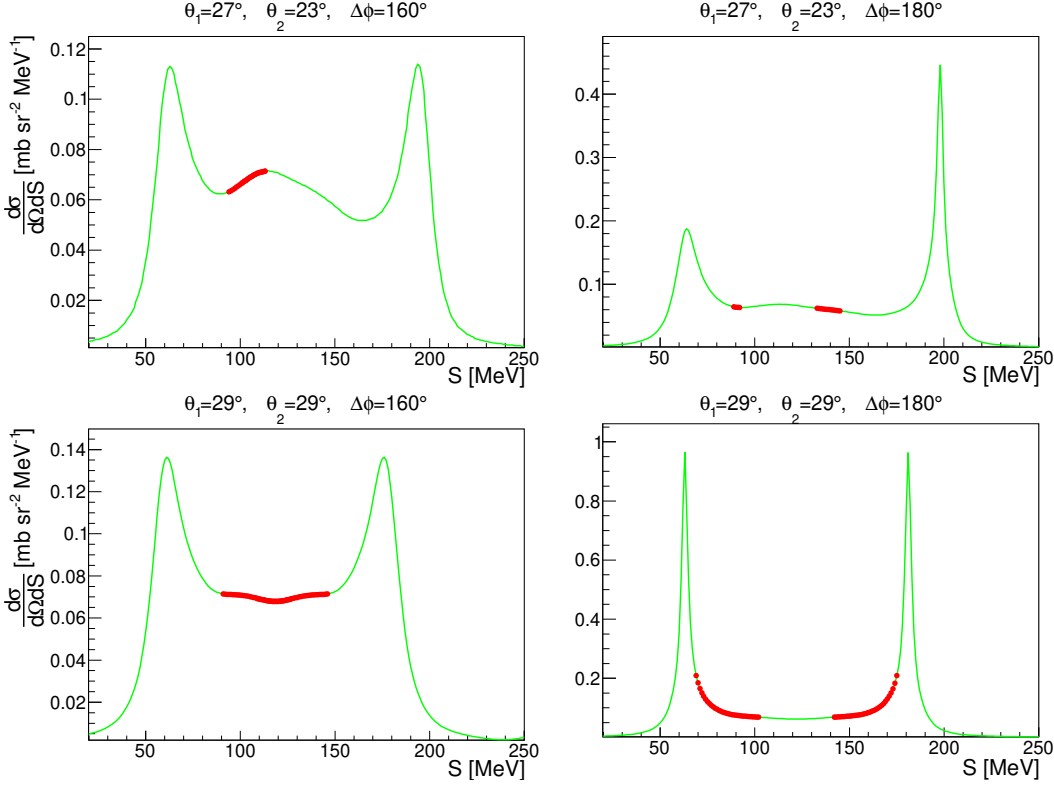

Figure 4: Examples of the differential cross sections of $^1\text{H}(d,pp)n$ breakup for two symmetric and two asymmetric angular configurations, specified in the figure. Green lines refers to the theoretical calculations based on the CD Bonn potential. Red solid lines and dots represent the regions useful for the cross-check analysis.

**Funding information** This work was supported by the Polish National Science Center under Grants No. 2012/05/E/ST2/02313, No. 2012/05/B/ST2/02556, No. 2016/22/M/ST2/00173 and No. 2016/21/D/ST2/01173.

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
