# Peer review of "Determination of phase space region for cross-check validation of the neutron detection in the BINA experiments"

_SciPost Physics Proceedings, doi:SciPost Phys. Proc. 3, 006 (2020)_

## Round 1 · Referee Report · Anonymous (Referee 1) · 2019-11-24

Strengths

  1. This paper addresses experimental measurements of a fundamental few-body process
  2. The technique that is described is to ensure an accurate calibration of an important piece of apparatus
  3. The technique for calibration is novel and well described.

Weaknesses

  1. The work is at a quite preliminary stage and follow-up work is required, but it is a worthwhile step in advancement to report at this stage.

Report

This is an experimental paper concerning the accurate calibration of apparatus for measuring the breakup of deuterons in d+p or d+d scattering at 160 MeV. The apparatus is installed at KVI Groningen. A forward wall of detectors can record both protons and neutrons and the more backward angles are also covered almost completely by a ball of detectors for protons. The paper concerns kinematic calculations to identify and quantify the cross section for those events in which all three particles from d+p going to p+p+n are recorded in the detector array. The kinematic completeness then means that for these events the neutron detection can be accurately calibrated from a comparison between the experimental measurements and the values calculated from the two accurate proton measurements.

Requested changes

There are a few typographical/grammatical changes that would improve the paper, but the readability is already excellent: 1. Last para of section 1: "A new generation of experiments..." and just before that, "like the coupled channel approach". 2. More than typographical but not critical: it would be good in the experimental setup to mention the thicknesses of the "thin" and "thick" scintillators. 3. Figure 1 caption, replace "adopted" by either "adapted" or "taken", whichever is accurate. 4. Section 3 first paragraph "...efficiency for neutron detection, estimated..." 5. After eq.1: "Examples of kinematics associated with different angular selections are..." 6. Following paragraph: "the following procedure has been developed" 7. Same paragraph: "and its energy exceeds..." and "Having these regions identified.." 8. It would be good to list the approximate thresholds for each type of particle. 9. Figure 3 caption: after "angles." add "(typical values for these angles are shown in fig. 4)" or similar. 10. In the conclusion: "possibility to extract the cross section", and "will be applied to the data", also "systematic uncertainties of the most important" 11. I suggest replacing "tools" after the above item 10 by either "variables" or "quantities".

---

## Round 2 · List of Changes

1. Change of affiliation of one of the authors, M. Eslami-Kalantari from KVI to Yazd university.
As referee suggested:
2. "like coupled-channel approach" to "like the coupled channel approach".
3. Adding the thickness of the scintillators.
4. Figure 1 caption, replacing "adopted" by "adapted".
5. Replacing "efficiency for the neutron" by "efficiency for neutron".
6. After eq.1: "Examples of kinematics associated with different angular selections are...".
7. Page 4: "procedure have been" - "procedure has been".
8. Same paragraph: "exceed" by "exceeds" and "this" by "these".
9. Adding value of the energy thresholds.
10. Figure 3 caption: after "angles." "(typical values for these angles are shown in fig. 4)".
11. Last paragraph: "extract cross section" by "extract the cross section".
12. "applied for" by "applied to".
13."of most" by "of the most".
14. Replacing "tools" by "quantities".

---

## Editorial Decision

published